# Analysis on single nucleotide polymorphisms of the *PeTPS-(-)Apin* gene in *Pinus elliottii*

**Lei Lei**, **Lu Zhang***, **Junhuo Cai, Min Yi, Heng Zhao, Jikai Ma, Meng Lai, Cangfu Jin**

Jiangxi Key Laboratory of Silviculture, 2011 Co-Innovation Center of Jiangxi Typical Trees Cultivation and Utilization, College of Forestry, Jiangxi Agricultural University, Nanchang, China

* zhanglu856@mail.jxau.edu.cn

## Abstract

### Background

Resin-tapping forests of slash pine (*Pinus elliottii*) have been set up across Southern China owing to their high production and good resin quality, which has led to the rapid growth of the resin industry. In this study, we aimed to identify molecular markers associated with resin traits in pine trees, which may help develop marker-assisted selection (MAS).

### Methods

*PeTPS-(-)Apin* gene was cloned by double primers (external and internal). DnaSP V4.0 software was used to evaluate genetic diversity and linkage disequilibrium. SHEsis was used for haplotype analysis. SPSS was used for ANOVA and $\chi^2$ test. DnaSP v4.0 software was used to evaluate genetic diversity.

### Results

The full length *PeTPS-(-)Apin* gene was characterized and shown to have 4638 bp, coding for a 629-amino acid protein. A total of 72 single nucleotide polymorphism (SNP) loci were found. Three SNPs (CG615, AT641 and AG3859) were significantly correlated with α-pinene content, with a contribution rate > 10%. These SNPs were used to select *P. elliottii* with high α-pinene content, and a 118.0% realistic gain was obtained.

### Conclusions

The *PeTPS-(-)Apin* gene is not uniquely decisive for selection of plus slash pines with stable production, high yield, and good quality, but it can be used as a reference for selection of other resin-producing pines and other resin components.

## Introduction

Resin, synthesized in the trunk of a pine tree, is a mixture of terpenoids. It is distilled to produce liquid turpentine and solid resin. Turpentine is a volatile essential oil, mainly a mixture

**Data Availability Statement:** All relevant data are within the paper and its Supporting Information files.

**Funding:** The work was funded by the National Key R&D Program of China (2017YFD0600502-5) and

the Research Project of Jiangxi Provincial Department of Forestry (No.201811).

**Competing interests:** The authors have declared that no competing interests exist.

of monoterpenes and semiterpenes. It can be directly used to dilute oil paints and analgesics and can also be processed into camphor, peppermint, terpineol, and spices. Resin is a translucent solid and a mixture of various diterpenoids. It can be used as a raw material in the industrial production of paints, rubber, inks, adhesives, dyes, and coatings [1]. With the introduction of slash pine (*Pinus elliottii*) from the United States, the available raw material resources of resin have been greatly increased. Previously, 90% of resin in the Chinese market was obtained from masson pine (*P. massoniana*) and a small amount from Simao (*P. kesiya*) and Yunnan (*P. yunnanensis*) pines, which are native trees. We found that the average annual resin yield of a slash pine individual is 5.0 kg, which is the highest resin yield among resin-producing-pines species. In slash pine, the income from resin far exceeds that from wood [2, 3].

The chemical composition of resin has a substantial influence on its quality, and different components have different uses in industrial production. Research has shown that the chemical components and contents of resin are heritable and controlled by additive and dominant gene effects [4]. Li (2012) selected four families with high turpentine content from 49 half-sibling families of slash pine as seed orchard materials, and the realistic genetic gain of turpentine content reached 426.15% [5]. Our group (2015) selected seven individuals with high turpentine content from previously selected 186 plants with high resin content and obtained a genetic gain of 51.57% in turpentine content based on high resin content [2].

However, these breeding efforts are based on mature forests, and we hoped to solve the problem of long generation cycles of trees using molecular biology methods. The way to solve this problem is to identify relevant molecular markers for early selection through association analysis. The methods of association analysis mainly include genome-wide association studies (GWAS) and association analysis of candidate genes. The pine tree, however, has a large genome, more than seven times the size of the human genome, with nearly 80% repeat sequences [6]. Even though the whole genome of loblolly pine (*P. taeda*, the model organism of pine species) has been sequenced, we were still unable to conduct GWAS studies in a short time [7, 8]. We can only rely on other method—candidate gene association analysis, which assumes candidate genes according to physiological functions and biochemical processes associated with the target trait [9]. Currently, association analysis studies of pine trees mainly focus on functional candidate gene strategies labeled with SNPs [10, 11]. Candidate genes associated with resin traits have been cloned from resin-producing-pines trees such as *P. densata* [10, 12], *P. densiflora* [13], *P. taeda* [14], *P. elliottii* [15], *P. kesiya var.langbianensis* [16], and *P. massoniana* [17]. There are also a few reports on the association between these genes and pine resin traits [11]. However, there are no reports in which these genes have been used to improve the pine tree itself. This may be related to the complexity of the resin trait, which is a quantitative character. The improvement in production capacity and resin quality using molecular biological methods should be an important method for pine trees.

Terpene synthase (TPS) is one of the key enzymes in the biosynthesis of the resin mixture and is also the enzyme found in most species and with the most diversified functions. TPS can be divided into six gene families: *Tpsa*, *Tpsb*, *Tpsc*, *Tpsd*, *Tpse*, and *Tpsf*. *Tpsd* is a unique gene family in gymnosperms, consisting of three subfamilies: monoterpene synthase gene *Tps-d1*, sesquiterpene thin synthase gene *Tps-d2*, and diterpene synthase gene *Tps-d3* [18]. Different plants contain the *TPS* gene, which encodes proteins with highly similar amino acid sequences. Although the genes probably evolved from a common ancestor, the distribution of exons and introns and their quantities are not the same among species [19]. It is generally believed that intron loss occurred during TPS gene evolution, that is, the shorter the length or the lower the number of introns, the higher the degree of evolution. The *TPSd* gene families in gymnosperms have more number of and longer introns than the *TPS* gene families in angiosperms [19]. These studies provide evidence for the cloning and expression of *TPS* gene in *P. elliottii*.

α-Pinene, a monoterpene compound with the highest content in turpentine mixtures, is a critical resistance molecule against insect, bacterial, and mechanical trauma in pines. It has left- and right-handed forms but is generally present in the left-handed form in resin [20]. The left handed α-pinene synthetase gene (*(-)Apin*) has been cloned in *P. taeda* [14], *P. contorta* [21], and *P. banksiana* [21]. Studies show that *(-)Apin* gene expression in pine trunk is correlated with α-pinene content in turpentine [22], which in turn is positively correlated with the production of turpentine and is positively or negatively correlated with other turpentine components. The identification of molecular markers related to high yield, superior quality resin would significantly promote genetic improvement and germplasm innovation. Compared with linkage analysis, correlation analysis can identify important alleles that are closely related to phenotypic traits.

To search for molecular markers related to resin traits to be used in marker-assisted selection (MAS), the *PeTPS-(-)Apin* gene was cloned, and its single nucleotide polymorphisms (SNPs) and linkage disequilibrium were analyzed. Functional SNPs were screened by ANOVA, 110 samples were typed, and the optimal selection scheme of *P. elliottii* with high α-pinene content was also considered. Selection of valuable plus trees will provide excellent materials for the breeding of resin-producing-pines and also provide a theoretical basis for the sustainable development of the resin market.

## Materials and methods

### Plant materials

Progeny tests were conducted at Baiyunshan forest farm (26°51′N, 115°11′E), Ji'an City, Jiangxi Province, China. The slash pines were cultivated on a flat woodland (hilly red soil, subtropical climate, 90 m altitude, 1646 mm rainfall, average temperature of 18.6°C). Three progeny trials comprised 110 open-pollinated families of superior trees that were collected from three seed orchards in the United States and one in China (listed in Table 1) and then planted in the spring of 1990 at Ganzhou, Ji'an, and Jingdezhen in Jiangxi Province, Southern China. The progeny trial had a completely randomized block design with five replicates, each consisting of a four-tree plot, and the distance between each tree was 4 m (Figs 1 and 2). A single plant of each family was randomly selected in the II block group of the test forest. If all the plants were missing (those suffering from damage and pests), a single plant of this family was randomly selected in the other block group. A total of 110 samples from 110 families were obtained.

### *PeTPS-(-)Apin* gene cloning

Prior to gene cloning, the tender buds or needles of *P. elliottii* were collected, labeled, and then stored at -80°C. DNA was extracted using the TIANGEN KIT and diluted to 50 ng/μL. The working solution was kept at 4°C, and the stock solution was kept at -80°C. *PeTPS-(-)Apin* gene was obtained from genomic DNA using double outer and inner primers (sequences are listed in Table 2). PCR amplification was performed using the PrimeSTAR HS DNA

**Table 1. Basic information of test materials.**

| Group | Sources of Sample Trees | Number of Families |
|:-----:|:------------------------|:------------------:|
| **A** | Seed orchards of Georgia, USA | 12 |
| **B** | Seed orchards of Mississippi, USA | 48 |
| **C** | Seed orchards of Florida, USA | 45 |
| **D** | Forestry Research Institute of Ji'an, China | 5 |

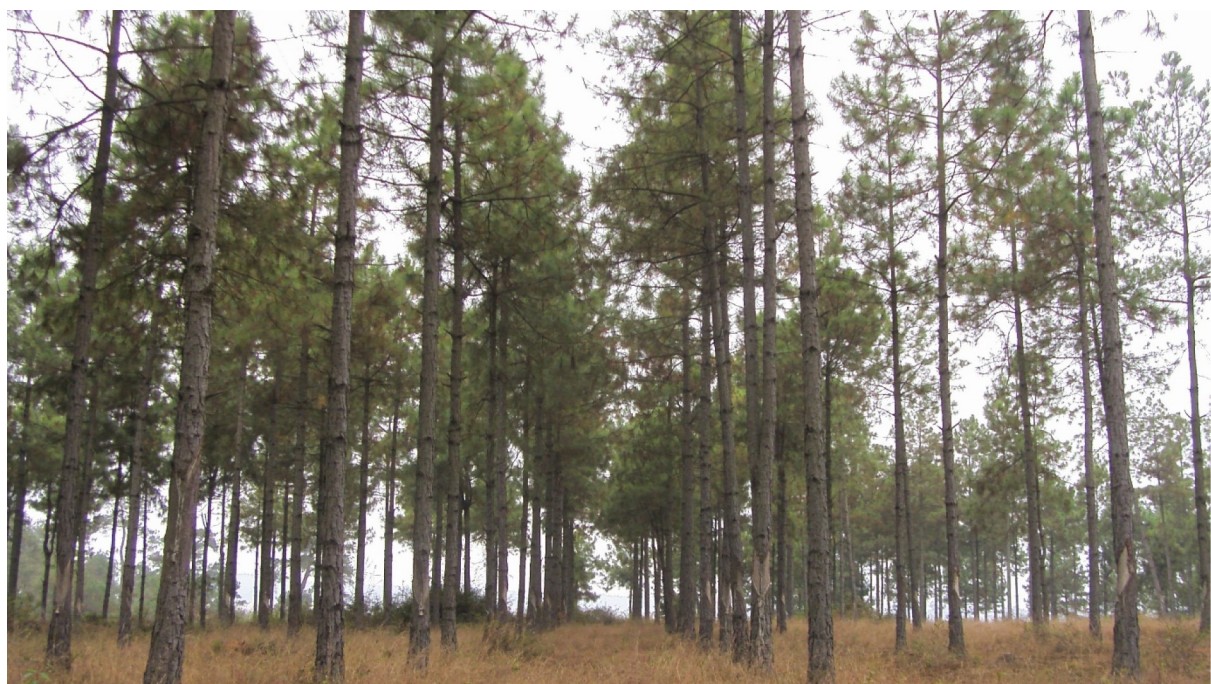

**Fig 1. Photo of test forest.**

Polymerase (Takara, Japan). The reaction volume in the first round of amplification (outer primes) was 10 µL, and in the second round of amplification (inner primes) 50 µL. PCR products were sequenced by the Shanghai Sango company.

## SNP site analysis and typing

Sequences were analyzed using Chromas 2.3 to contrast the peaks, and Excel software to organize the SNP typing results in an A, T, C, and G format. Insertion/deletion mutations were

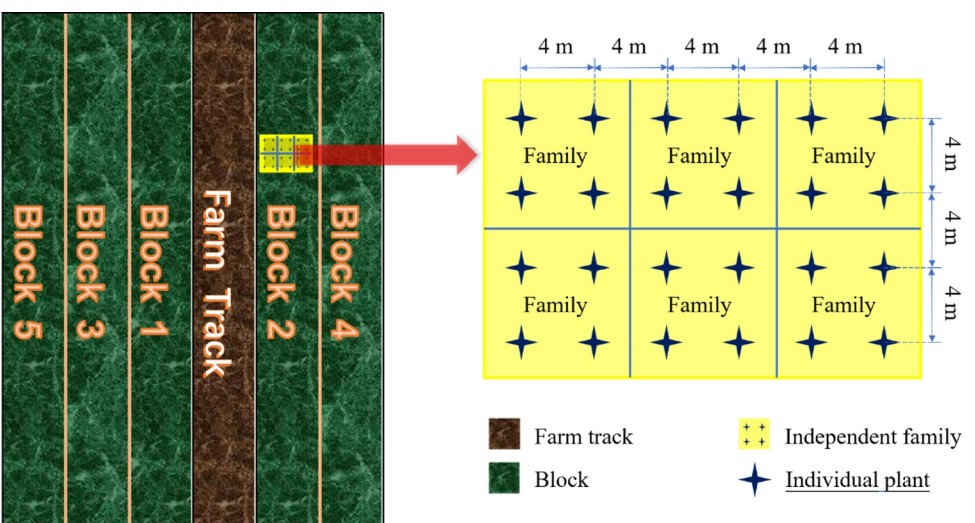

**Fig 2. Schematic diagram of test forest.**

**Table 2. Primer sequences for *PeTPS-(-)Apin* gene cloning.**

| Primer | Sequences (5'~3') |
|---|---|
| **outer-F** | ATAGTCCTTGAATTGTGGAG |
| **outer-R** | GAGAAAAATCCTACTGGTGT |
| **inner-F** | GAGACGTATTGCGATGTTAT |
| **inner-R** | CCAAATGTATTGATTGAGGG |

removed from the combined gene information file, and the results were imported into DnaSP 4.0 software, which was used to calculate basic SNP information such as number of inversions and conversions, frequency, nucleotide polymorphism ($\pi$, $\theta w$), synonymous mutation diversity, non-synonymous mutation diversity, silencing site diversity, and linkage disequilibrium. SHEsis was used for haplotype analysis. ANOVA and $\chi^2$ were performed to evaluate if the results conformed to the Hardy-Weinberg equilibrium using SPSS software.

## Results

### *PeTPS-(-)Apin* gene sequence

After cloning, sequencing, and assembly, the full length sequence of the *PeTPS-(-)Apin* gene was obtained. The genome was of 4638 bp and included 10 exons and 9 introns, and encoded a 629 amino acid protein (Fig 3). Its homology with *Pt1* (gb|AF543527.1|), *PcTPS-(-)apin1* (gb|JQ240303.1|), *PmTPS-(-)apin* (gb|KF547035.1|), and bankerson *PbTPS-(-)apin1* (gb|JQ240304.1|) was 91–99%, with three long conserved regions (> 50 a.a.). Homology of *PeTPS-(-)Apin* was 81–87% to north American spruce *PsTPS-Pin* (gb|AY237645.1|) and European spruce *PaTPS-Pin* (gb|AY473622.1|), with five short conserved regions (> 15 a.a.). It had a homology of 73–79% with fir *Ag3.18* (gb| U87909.1|), *PgQ028* (gb| BT105745.1|), and *PgWS00725* (gb| HQ426153.1|), with two shor,t conserved regions (> 15 a.a.).

### Diversity analysis of SNPs

After removing a 46 bp insertion/deletion mutation, the 4592 bp sequence was used for further analysis. A total of 72 SNP loci were observed, with 37 transitions and 35 transversions, and 59 SNP loci were of high information content. The average and high information content SNP frequencies were 1/64.42 and 1/78.61, respectively, with 47 silent mutation sites and 12 non-identical mutation sites. The nucleotide polymorphism of $\pi$ 0.00276 and $\theta w$ 0.00259, were at a low level, probably because more polymorphisms were not detected form the small sample size in this study. The diversity of silencing loci of 0.00316, diversity of synonymous mutations of 0.0009, and diversity of non-synonymous mutations of 0.00202 were all at a low level. These

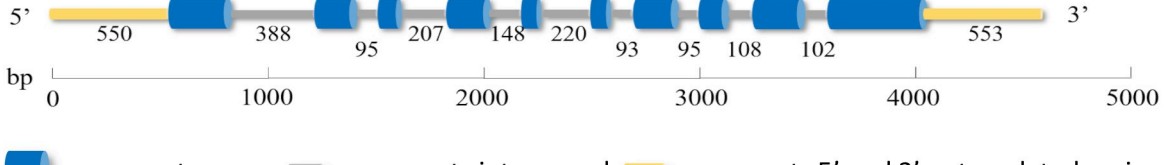

represents exon, represents intron, and represents 5' and 3' untranslated region.

The length of the exons is marked on top, and that of the introns blow, in bp.

**Fig 3. Schematic representation of the *PeTPS-(-)Apin* gene structure.**

results indicated that the nucleotide sequence of this gene is highly conserved. Its haplotype diversity was 0.897, its non-synonymous mutation frequency in the coding region (Ka) was 0.00202, its synonymous mutation frequency in the coding region (Ks) was 0.00317, its minimum number of historical recombination events (Rm) was 4, and its Rm/SNPs was 0.068. In addition, Ka/Ks was 0.6382, which, being less than 1, suggested that the gene was undergoing balanced selection. Furthermore, three methods (Tajiama D, Fu-Li, Fay, and Wu's H test) were used to detect neutral selection, and the results were 0.23743, 0.20077, and 0.1318, respectively. All three values were greater than 0 (positive), but the test values seldom reached a sufficiently significant level to indicate that this gene followed a neutral model in evolution and is undergoing balanced selection.

## Linkage disequilibrium

The level of linkage disequilibrium varied greatly in different species and different gene sequences. A correlation analysis was established based on linkage disequilibrium (LD), and different LD levels determined different strategies for correlation analysis. An LD attenuation diagram of 59 highly informative SNP loci of the *PeTPS-(-)Apin* gene is shown in Fig 4. As a result, the $r^2$ value decreased to 0.2 at approximately 1000 bp and below 0.1 at approximately 2000 bp, indicating a high level of LD and a close linkage between these markers. To better understand the linkage relationship among SNPs, an LD matrix diagram of the gene was constructed, with darker color representing greater connection (Fig 5).

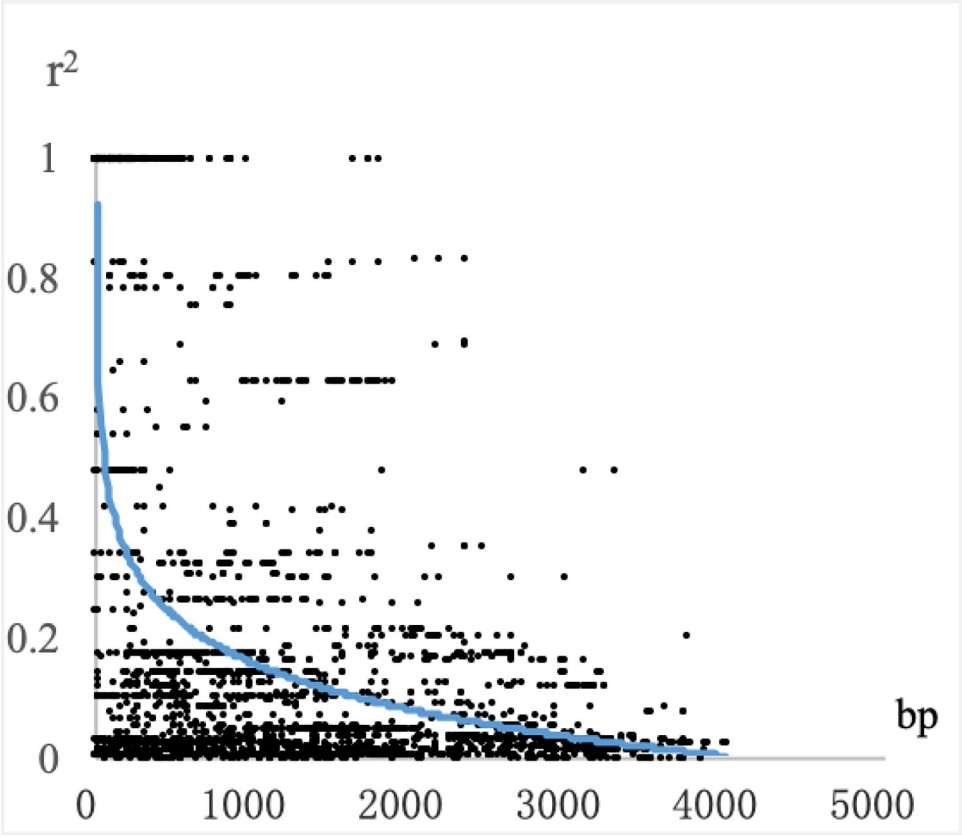

**Fig 4. LD attenuation diagram of *PeTPS-(-)Apin*.**

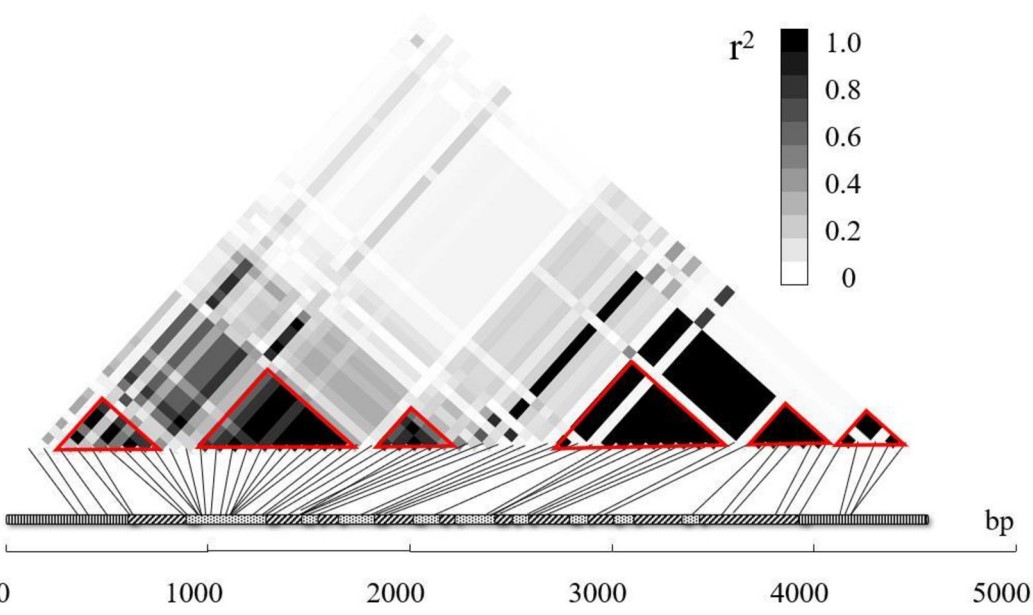

The darker the color, the more connected it is. ▨ as exon, ▨ as intron ▨ as 5'

and 3' untranslated region, and △ as haplotype blocks.

**Fig 5. LD matrix diagram of the *PeTPS-(-)Apin* gene.**

## Haplotype block

Owing to the high LD level of the *PeTPS-(-)Apin* gene, and the close linkage between SNP sites, we firstly separated the 59 SNP sites after removing 12 rare SNPS (MAF < 0.05) into six haplotype blocks (named (-)Apin-1, (-)Apin-2, (-)Apin-3, (-)Apin-4, (-)Apin-5 and (-)Apin-6). Haplotype analysis was performed based on $r^2$ value. All the 110 samples were used for haplotyping, and the length of the haplotype blocks was 150–1400 bp. Theoretically, the number of haplotypes should be twice that of the number of SNPs contained in the block, but the actual number of observed haplotypes scarcely approached the theoretical value. This may result from the close linkage between SNP loci. Additionally, the frequency distribution of genotypes of different haplotype blocks was varied, and there were 1–3 dominant genotypes in each haplotype (frequencies greater than 0.2). An overview of the haplotype blocks is shown in Table 3.

## Genotyping of SNPs association with resin traits

A total of 29 resin traits, including two related to resin-producing capacity and 27 resin components, were examined. The traits related to resin-producing capacity included basic resin-producing capacity ($W_0$), and potential resin-producing capacity ($W_P$). The resin components included turpentine, resin, 7 kinds of turpentine, and 18 kinds of resin. The seven types of turpentine components were α-pinene, camphene, β-pinene, dipentene, cymene, myrcene, and cycloisilongifolene. The 18 types of resin components were pimanthrene, pimarinal, pimaric acid, elliotinoic acid, sandaracopimaric acid, dehydroabietic aldehyde, isopimaric acid, levopimaric acid, palustric acid, 6,8,11,13-abietatetraenoic acid, dehydroabietic acid, abietic acid, neoabietic acid, mercusic acid, 7,13,15-abietatrienoic acid, 8,14-dihydro pimaric acid, 15-hydroxyl hydrogen abietic acid, and 7-hydroxyl hydrogen abietic acid. The determination methods of these resin traits were shown in S1 Text and S1 Table. The variation statistics of

**Table 3. *PeTPS-(-)Apin* gene haplotype blocks of the 110 samples.**

| Haplotype | Location | Length (≈bp) | Number of SNPs | Number of haplotypes | Dominant haplotype block | Frequency |
|---|---|---|---|---|---|---|
| (-)**Apin-1** | 5'-UTR, 1ex | 600 | 11 | 11 | AACCAGGTCGA | 0.636 |
| (-)**Apin-2** | 1in | 150 | 9 | 16 | AAGGTCAAA | 0.554 |
| (-)**Apin-3** | 2-3ex | 500 | 10 | 12 | CAACGCGAGA | 0.346 |
| | | | | | TAATTCGAGA | 0.218 |
| | | | | | TAGTTCGAGG | 0.255 |
| (-)**Apin-4** | 4-5ex | 1000 | 10 | 15 | CTGCCAAGGA | 0.300 |
| | | | | | TATTGGAGTT | 0.291 |
| (-)**Apin-5** | 6-10ex | 1400 | 12 | 11 | TAGTGTTCGCAT | 0.364 |
| | | | | | ATAGTGTTCTAT | 0.291 |
| | | | | | CGCCGTCCAGAT | 0.200 |
| (-)**Apin-6** | 3'-UTR | 300 | 7 | 12 | ACTCCC | 0.355 |
| | | | | | TACCCC | 0.236 |

Haplotypes with frequencies greater than 0.2 were considered dominant.

the 29 resin characters is shown in S1 Schedule. The correlation between 59 SNP loci of the *PeTPS-(-)Apin* gene and 29 types of resin traits was analyzed. Three SNPs (CG615, AT641 and AG3859) were associated with the phenotypic trait of α-pinene content, and with a higher contribution rate ($R^2 \geq 10\%$) (Table 4). Genotyping and polymorphism analysis were performed on 110 samples of *Pinus elliottii*. The expected heterozygosity, had an average of 0.5365, and the observed heterozygosity, had an average of 0.5030 (Table 5). Therefore, *PeTPS-(-)Apin* is a candidate gene for positive tree screening with high α -pinene content.

## Discussion

Three SNPs (CG615, AT641 and AG3859) were selected as TagSNPs related to α-pinene content. Among them, mutations in AG3859 may lead to increased activity of some enzymes regulating the synthesis of α-pinene. In this study, 12 plus trees were selected, and the actual gain of α-pinene content was increased by 44.39% without decreasing the contents of $W_O$, $W_P$, turpentine or β-pinene content.

### Analysis of the TagSNPs

As the HapMap project progressed, a large amount of SNP site information accumulated in the human genome database, and tagging SNPs (i.e., TagSNPs) need to be screened from these data to reduce interference from redundant data for more precise location of disease-associated SNP sites [23]. The subject of this work was slash pine (*P. elliottii*), a less studied biological species, for which there is still very little SNP data [24]. Loblolly pine (*P. taeda*) is the most thoroughly studied pine species and its complete genome sequence is available, but repeat sequences hamper the expansion of the SNP database. There is no GWAS report of related species in the short term [8]. Moreover, as a complex quantitative trait, pine resin trait needs a

**Table 4. Overview of SNPs association with α-pinene content.**

| SNPs | Haplotype blocks | Effect R2 (%) |
|---|---|---|
| **CG615** | (-)Apin-1 | 15.548 |
| **AT641** | (-)Apin-1 | 10.271 |
| **AG3859** | (-)Apin-5 | 11.445 |

**Table 5. Genotypes frequency and polymorphisms of 110 *Pinus elliottii* samples based on 3 SNPs.**

| SNP | Genotype frequency | | | Allele frequency | | $H_e$ | $H_o$ | $\chi^2$ | HWE |
|---|---|---|---|---|---|---|---|---|---|
| | G | N | F | A | F | | | | |
| CG615 | CC | 15 | 0.1364 | C | 0.5909 | 0.5489 | 0.5091 | 0.0351 | 0.8514 |
| | CG | 50 | 0.4545 | G | 0.8636 | | | | |
| | GG | 45 | 0.4091 | | | | | | |
| AT641 | AA | 15 | 0.1364 | A | 0.6000 | 0.5118 | 0.4636 | 0.0013 | 0.9710 |
| | AT | 51 | 0.4636 | T | 0.8636 | | | | |
| | TT | 44 | 0.4000 | | | | | | |
| AG3859 | AA | 23 | 0.2091 | A | 0.7455 | 0.5489 | 0.5364 | 0.6179 | 0.4318 |
| | AG | 59 | 0.5364 | G | 0.7909 | | | | |
| | GG | 28 | 0.2545 | | | | | | |

G, genotype; A, allele; N, number; F, frequency; $H_e$, expected heterozygosity; $H_o$, observed heterozygosity; $\chi^2$, chi-square test value; HWE, Hardy-Weinberg equilibrium (when the value > 0.05, it indicated a Hardy-Weinberg equilibrium; the data came from the same Mundell population).

large number of samples for its gene association studies. Most literature reports are of 200–400 individuals, and these individuals are naturally pollinated populations with distant genetic relationships [25].

In this study, however, we only analyzed 110 families introduced from the United States, because slash pine is not a native species. To ensure a distant genetic relationship between individuals, it is difficult to collect more than 200 individuals as association groups. Based on a small data volume of nucleotides database (less than 300), our existing SNP data cannot cover the genome-wide. Moreover, it is difficult to obtain more SNP data in the short term due to the enormous genome (>20,000 Mbp). Therefore, simple associations (ANOVA) were used for association analysis, which may be controversial. Nevertheless, *PeTPS-(-)Apin* gene was considered as an important candidate gene for α-pinene content, and three TagSNPs (CG615, AT641 and AG3859) were associated with α-pinene content.

## Molecular mechanism of mutations

Using candidate gene association analysis, we found three TagSNPs (CG615, AT641, and AG3859) that may be related to α-pinene content, and analyzed the molecular mechanisms of these three mutations (Fig 6). These three mutations were all non-synonymous mutations in exon 1 (haplotype block (-) APin-1) and exon 10 (haplotype block (-) APin-5): G at CG615 was mutated to C, resulting in a change from arginine, to proline; T at AT641 was mutated to A, changing phenylalanine, to tyrosine; and G at AG3859 was mutated to A, changing arginine to lysine. We used the tools ElM and CDD to predict the possible functional sites of *PeTPS-(-)Apin* gene: AG3859 is the first amino acid of the LIG_FHA ligand functional domain. This domain binds forkhead-associated (FHA) phosphopeptide ligands that consist of seven amino acids, the phosphopeptide identification domain of many regulatory proteins [26]. The mutation at AG3859 may lead to increased activity of some enzymes that regulate the synthesis of α-pinene. The functions of CG615 and AT641 have not yet been predicted, but we do know that the mutations are at the N-terminus of the protein, near the conserved domain of the TPS gene families, and are closely linked (amino acids are at positions 10 and 19, respectively). Whether the linkage of mutations and α-pinene content is an artifactual event caused by sample size and population structure, or a real event caused by natural selection, is unclear, and its molecular mechanism remains to be studied.

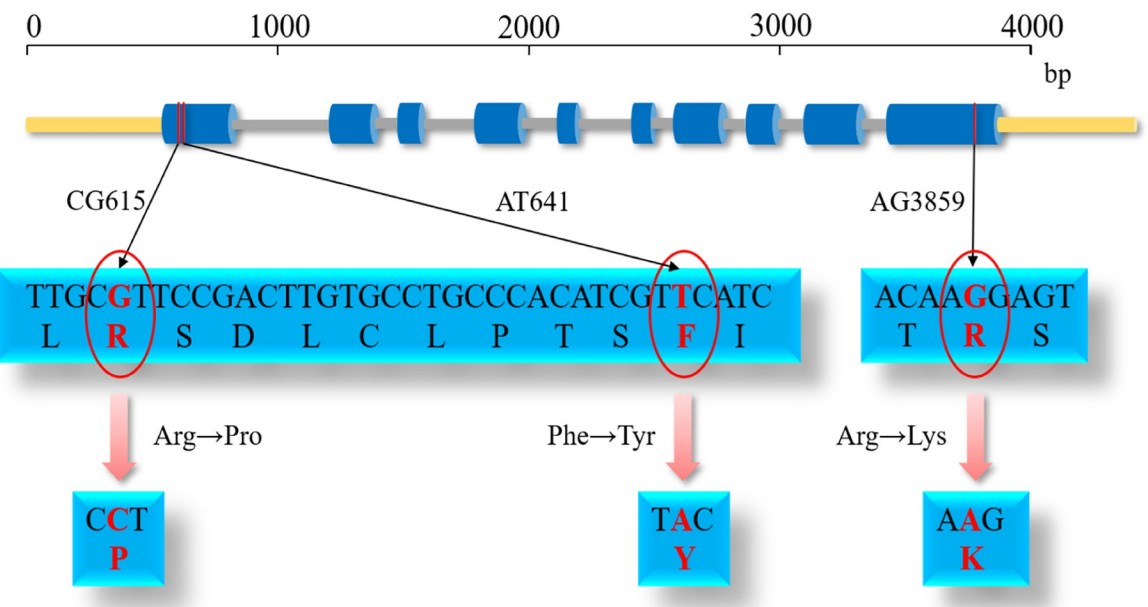

**Fig 6. Molecular mechanism underlying the effects of the three TagSNPs on α-pinene content.**

A large number of studies have shown that some traits can be controlled by a single gene (or even a single site mutation), such as that related to resistance. Although most complex traits are controlled by multiple genes, there may be dominant, additive, epistatic, and interaction effects among these genes, such as those that occur in tree height, weight, and yield. Studies have shown that the synthesis of α-pinene is controlled by a specific synthetase, that is, the internal mutation of *PeTPS-(-)Apin* gene may lead to more active functional sites, thus increasing the content of α-pinene. However, in actual production, we found that the content of α-pinene was affected by time and space, and therefore is a more complex quantitative trait. More candidate genes related to it need to be mined.

## Selection of plus trees

As the demand for processed resin products has exceeded supply, the prices of certain chemical components have rapidly increased each year. Therefore, the supply of industrial raw materials can only be guaranteed by increasing the content of specific components of the resin itself. There are significant differences between species, even of the same species, although the chemical composition of resin is similar. For example, in turpentine of *P. kesiya*, α-pinene accounts for over 90% of monoterpene content, β-pinene content is as high as 25.9%, and there is even a high content of Δ3-carene in rare breeds [27]. In the resin of *P. elliottii*, pimaric type acid content is 9.93% and that of isopimaric acid up to 7.6%, whereas isopimaric acid content in *P. massoniana* is less than 1% [28]. We suggest that the chemical composition and content should be considered important criteria of resin quality. Prior to evaluating the quality of resin, one should consider the content of the expensive ingredients that are widely used in industry [29]. Studies have shown that single components in resin are synthesized under the control of single genes, and its content is controlled by a gene that has a dominant effect [30]. These gene variants can be targeted for improvement through artificial selection. The heritability of main components of turpentine is from 0.2 to 0.6, and the α-pinene is 0.3–0.5 [2]. In

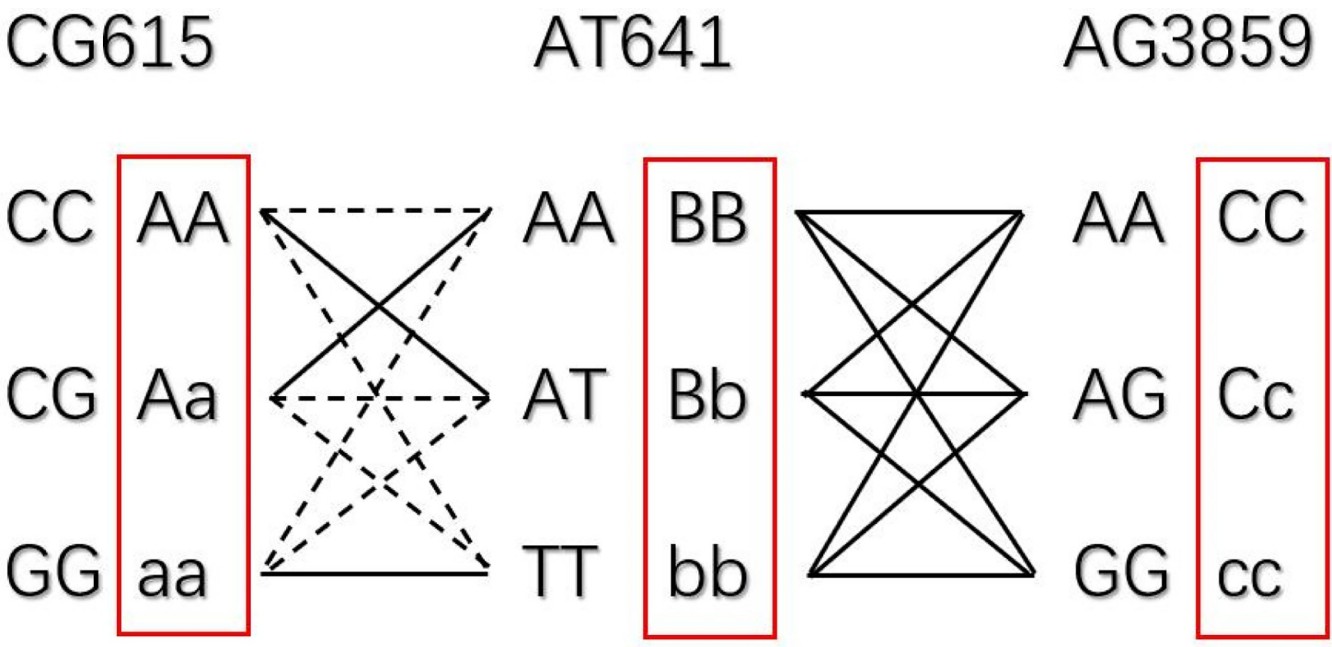

**Fig 7. Schematic diagram of genotypes.** Dotted lines indicate the possible genotypes that were not observed.

recent years, there has been an increasing number of directional selection and breeding programs of resin components in resin-producing-pines in China. The breeding targets involve α-pinene, β-pinene, Δ3-carene, dipentene, abietic acid, isopimaric acid, pimanthrene, and

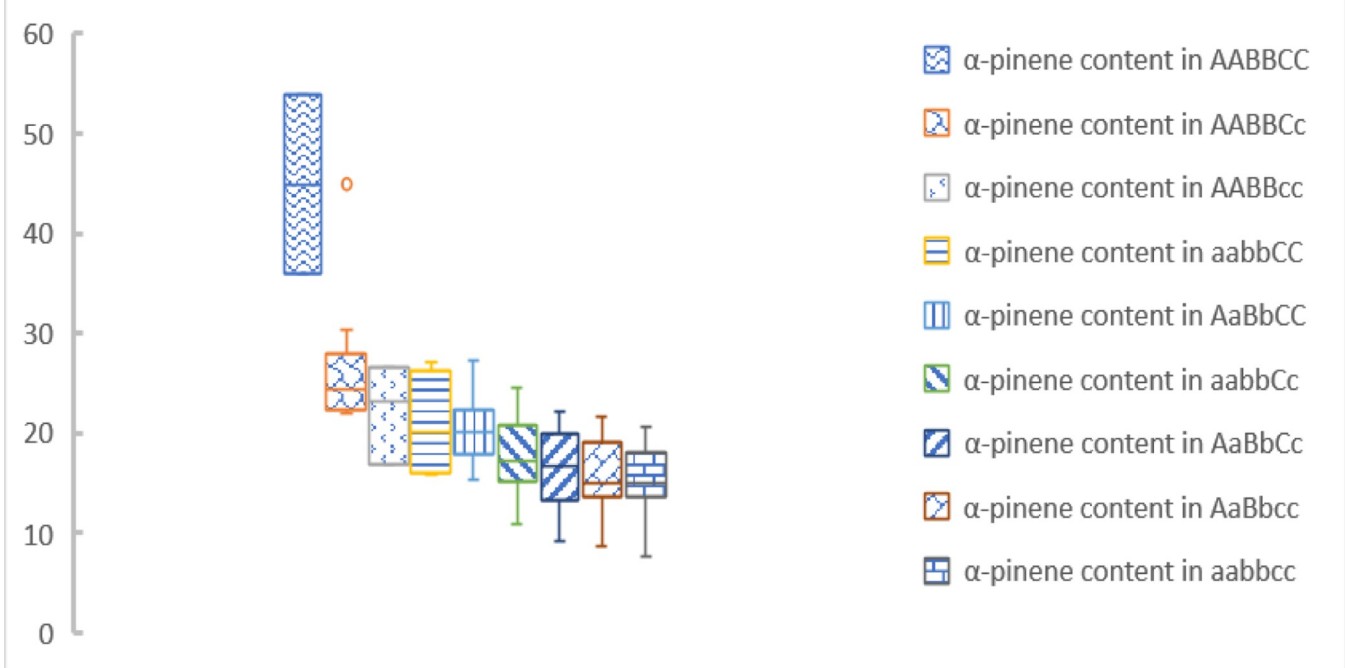

**Fig 8. Box map of α-pinene content for 9 genotypes.** Among them, the genotype AABBCC was only 1, and the box map was not drawn.

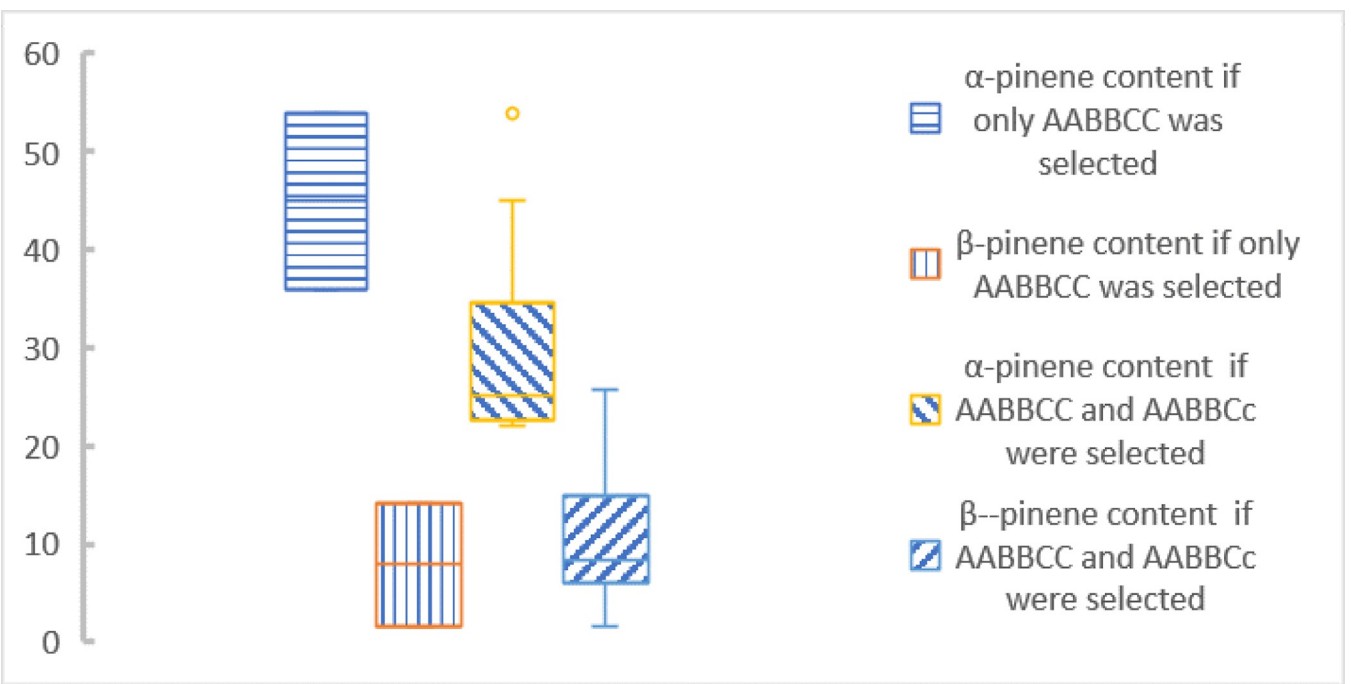

**Fig 9. Box map of α-pinene and β-pinene content in two different selection methods.**

pimaric acid) [2, 5]. The application of molecular markers in the directional selection of specific components has not yet been reported.

In this study, three TagSNPs (CG615, AT641 and AG3859) were used for genotyping (Fig 7) and were used to select *P. elliottii* with high α-pinene content. There were only 10 genotypes were observed (27 are possible in Fig 7). Six genotypes with high α-pinene content (over 20%) were observed: AaBbCC, AABBCC, AABBCc, AABBcc, aaBbCc, and aabbCC. Among them, the α-pinene content of AABBCC was over 40%. Based on only AABBCC selection, two trees were selected (S2 Schedule, Fig 8). With a selection ratio of 1.82% and a selection differential of 24.32, we achieved a real gain of 118.0% in α-pinene content. In this way, $W_0$, $W_P$, and turpentine were not reduced, but there an 8.97% reduction in β-pinene content. Multiple studies have shown a significantly negative correlation between α-pinene and β-pinene contents, which may be related to these two synthase genes and their regulatory genes. The genetic correlation coefficients ranged from 0.36 to 0.46 [1, 2, 5]. We also tried another selection method. The α-pinene content of AABBCc reached 26%, and it was also considered an excellent genotype. Based on the selection of these two genotypes (AABBCC and AABBCc), 12 trees were selected (Fig 9, S2 Schedule), the selection ratio was 10.91%, and the selection difference was 9.15. This can increase the real gain of α-pinene content by 44.39% without reducing $W_0$, $W_P$, turpentine, or β-pinene content. In actual industrial production, β-pinene is also an important material, and we can choose the scheme according to different breeding objectives.

## Conclusions

Above all, the yield and quality of resin are complex quantitative traits. *PeTPS-(-)Apin* gene variants can be used to select *P. elliottii* trees with high α-pinene content. Although selection of plus *P. elliottii* for stable production, high yield, and quality resin with high α-pinene content is not only contingent on the *PeTPS-(-)Apin* gene, this gene can certainly be used as a

reference for selection of high yield pines and other components. However, further studies are needed.

## Supporting information

**S1 Fig. High definition view of Fig 1.**
(JPG)

**S2 Fig. High definition view of Fig 1.**
(JPG)

**S1 Text. Determination method for resin traits.**
(DOCX)

**S2 Text. Group structure analysis method and results.**
(DOCX)

**S3 Text. Full length sequence of *PeTPS-(-)Apin* gene.**
(DOCX)

**S4 Text. Original data of Figs 4 and 5.**
(DOCX)

**S1 Table. Original data of resin traits.**
(XLSX)

**S2 Table. Original data of Tables 4 and 5.**
(XLSX)

**S3 Table. Original data of Figs 7–9.**
(XLSX)

**S1 Schedule. Variation in 29 resin traits of 110 slash pines.**
(DOCX)

**S2 Schedule. Overview of 12 plus trees with high α-pinene content.**
(DOCX)

## Author Contributions

**Data curation:** Lei Lei.

**Formal analysis:** Lei Lei.

**Funding acquisition:** Lu Zhang.

**Investigation:** Lei Lei, Heng Zhao, Meng Lai.

**Methodology:** Lei Lei, Heng Zhao, Meng Lai.

**Project administration:** Lu Zhang.

**Resources:** Lu Zhang.

**Software:** Lei Lei, Min Yi.

**Validation:** Lei Lei, Min Yi.

**Writing – original draft:** Lei Lei.

**Writing – review & editing:** Junhuo Cai, Jikai Ma, Cangfu Jin.

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
