## [Decision Letter · Decision Letter 0]

5 Jun 2021

PONE-D-21-06793

Association of single nucleotide polymorphisms in the PeTPS-(-)Apin gene with resin traits in Pinus elliottii

PLOS ONE

Dear Dr. Zhang,

Thank you for submitting your manuscript to PLOS ONE. After careful consideration, we feel that it has merit but does not fully meet PLOS ONE’s publication criteria as it currently stands. Therefore, we invite you to submit a revised version of the manuscript that addresses the points raised during the review process.

We look forward to receiving your revised manuscript.

Kind regards,

Himanshu Sharma

Academic Editor

PLOS ONE

Journal Requirements:

Additional Editor Comments:

The manuscript entitled Association of single nucleotide polymorphisms in the PeTPS-(-)Apin gene with resin traits in Pinus elliottii has been thoroughly reviewed by the reviewers and they advised to revise the manuscript So I also stand with their decision so the authors revise each comment critically and resubmit the manuscript.

Reviewers' comments:

Reviewer's Responses to Questions

**Comments to the Author**

1. Is the manuscript technically sound, and do the data support the conclusions?

Reviewer #1: Partly

Reviewer #2: Yes

2. Has the statistical analysis been performed appropriately and rigorously? 

Reviewer #1: No

Reviewer #2: Yes

3. Have the authors made all data underlying the findings in their manuscript fully available?

Reviewer #1: Yes

Reviewer #2: Yes

4. Is the manuscript presented in an intelligible fashion and written in standard English?

Reviewer #1: Yes

Reviewer #2: Yes

5. Review Comments to the Author

Reviewer #1: In this study the authors aimed to identify molecular markers associated with 29 resin traits in Pinus elliotii, for their possible later use in marker-assisted selection. They use 110 trees and 120 SSRs to assess population structure, and then identify 72 SNPs in gene PeTPS-(-)Apin, related to rosin quality. They find evidence of 10 SNPs in this gene significantly associated with 7 resin traits, and suggest using the target gene for plus tree selection within a species’ breeding program.

Overall, this manuscript tackles an interesting topic by bringing together genetic and phenotypic data, for further tree breeding purposes. However, the manuscript is not properly structured and I think that several limitations of the analysis may compromise the findings. In its current state and in light of my major comments below, I do not think this manuscript is suitable for publication in PLOS ONE. However, if these issues below were to be comprehensively addressed, I would be happy to review a new version of the document.

Major comments:

1. The papers needs to be streamlined, and will benefit from rigorous scientific editing. While the english language is largely correct, the poor structure of the manuscript (intro, m&m, results, discussion, conclusions) makes it difficult to read and understand. Many unimportant details are provided along the text, while the most important ideas and hypothesis to be tested are not so much highlighted.

2. Regarding the methods, considering that this is a paper focused on breeding, I have missed important basic data, like for example heritability. Also genetic correlations among resin components. Are there correlations or trade-offs between the studied resin traits? I think it could be helpful to look for pairwise correlations between these traits. A classic breeding approach, including quantitative genetic data, would complement the understanding of the document, provided that the data are already available.

3. From the described methods, I have not found a justification for all the steps. Starting from the genetic structure analysis. A great number of SSRs (120) is used to assess population structure. But, how are the results taken into account in the discussion? For example: from the final 12 selected trees, do they come from different genetic pools? What can be done to widen the genetic basis of the breeding population? These are important questions regarding the sustainability of a breeding program.

4. However, my major concern is about the statistical methods used for association analysis. A said by authors, they cannot discard false positives because of the small population size, population structure and a reliable genetic distance matrix (L328-330). LD between markers and false positives due to kinship relations and population structure are indeed key issues to be solved in association analysis. First of all, I am concerned that the number (59) and distribution of SNPs used in this study may bias the results. The authors recognize that there is actually a high level of LD and close linkage between markers (L262-263, L278). This could certainly affect the results of the association test, increasing the number of associated markers. The number of trees (110) is also quite small for association testing. Regarding kinship, how where the genotyped trees selected within families? The K matrix was calculated using SSRs or SNPs? If that is the case, I would not rely too much on a K matrix calculated with only 59 SNPs. Regarding population structure, admixture coefficients would improve if averaged across replicated runs (using for example software CLUMPP). Moreover, after association analysis, multiple testing correction is also needed to control for false positives when a large number of traits are tested. One option could be using the false discovery rate (FDR) method (Storey and Tibshirani 2003). Overall, I have reasonable doubts that the associations described as significant are really significant, or could rather be false positives.

Minor comments:

- Introduction: I found excess literature about non-relevant information for the scope of the paper, such as information about resin production (L38-43, L50-52), while lacking key information about the genetic control of resin traits.

-Overall, the manuscript lacks bibliographic references. Examples: L58, 70, 83, 90, 258. I would encourage authors to upgrade their references with publications of high impact international journals.

-Materials and methods: regarding plant materials, it is not clear how the 110 trees genotyped and phenotyped for resin traits where selected from the 110 families of the progeny trials. In population structure analysis, I would clarify how the 0,1 format data was transformed to round numbers, corresponding to a bp position.

-L336-338: Why are SNPs in HWE discarded for further analysis and breeding pourposes?

-Discussion: a first short paragraph with main findings would be appreciated.

- Figure 1 should preferably belong to supplementary material.

- Figures 2 an 3 are redundant. They show the same information, in different scale. It would be useful to see how are progeny trial populations or families distributed in the x-axis.

- Suggestion for associated SNP data figure: Instead of table, a boxplot for each SNP marker, showing median and interquantile range of phenotype variation would be more visual and informative for the reader.

Reviewer #2: Manuscript may be considered after addressing the following comments, importantly point 2.

1. Provide details and references fr the methods used such as GC and MS procedures.

2. Attach a picture of cultivated plants (field trial) and plots made along with distances between each plants on which genetic analysis was performed. This is the core of the study.

3. Verify the data in Table 3. example red group totals 16.

4. Provide a heading Figure legends.

5. Some figures lacks legends.

6. Figure 5 needs more details about intron and exons or put it in the legend.

7. Mention the details of structure parameters used. Ex. Burn-in period, MCMC etc.

6. PLOS authors have the option to publish the peer review history of their article (what does this mean?). If published, this will include your full peer review and any attached files.

Reviewer #1: No

Reviewer #2: No

---

## [Author Response · Author response to Decision Letter 0]

12 Jul 2021

I have received your email, and have modified it one by one according to reviewer and editor comments , and uploaded it to the submission system.Thanks a lot.

---

## [Decision Letter · Decision Letter 1]

12 Aug 2021

PONE-D-21-06793R1

Association of single nucleotide polymorphisms in the PeTPS-(-)Apin gene with resin traits in Pinus elliottii

PLOS ONE

Dear Dr. Zhang,

Thank you for submitting your manuscript to PLOS ONE. After careful consideration, we feel that it has merit but does not fully meet PLOS ONE’s publication criteria as it currently stands. Therefore, we invite you to submit a revised version of the manuscript that addresses the points raised during the review process.

We look forward to receiving your revised manuscript.

Kind regards,

Himanshu Sharma

Academic Editor

PLOS ONE

Journal Requirements:

Additional Editor Comments (if provided):

Dear Authors I have received the reports from experts in the field on your manuscript one reviewer is still not agree for possible publication of the manuscript in the Plosone. So based on the decision I have to go with reviewer. and again manuscript has to be revised thoroughly. I am attaching comments which have to be revised.

Reviewers' comments:

Reviewer's Responses to Questions

**Comments to the Author**

1. If the authors have adequately addressed your comments raised in a previous round of review and you feel that this manuscript is now acceptable for publication, you may indicate that here to bypass the “Comments to the Author” section, enter your conflict of interest statement in the “Confidential to Editor” section, and submit your "Accept" recommendation.

Reviewer #1: (No Response)

Reviewer #2: All comments have been addressed

2. Is the manuscript technically sound, and do the data support the conclusions?

Reviewer #1: Partly

Reviewer #2: Yes

3. Has the statistical analysis been performed appropriately and rigorously? 

Reviewer #1: No

Reviewer #2: Yes

4. Have the authors made all data underlying the findings in their manuscript fully available?

Reviewer #1: Yes

Reviewer #2: Yes

5. Is the manuscript presented in an intelligible fashion and written in standard English?

Reviewer #1: Yes

Reviewer #2: Yes

6. Review Comments to the Author

Reviewer #1: Overall the authors tried to address most comments, improving the manuscript. However, I still think that statistical analysis may compromise their findings, and thus their results and conclusions. I’m especially concerned about association testing results, as highlighted in major comment 4. Thus, I do not think this manuscript is suitable for publication in PLOS ONE in its present form.

Response to authors for major comments review:

2. Authors now provide a citation about heritability for turpentine and alfa-pinene in the species. My sense was that there is a need for trustable genetic control if then you are going to perform an association test, which consists in stablishing an statistic relationship between fenotypes (SNPs) and genotypes.

My interest in testing for genetic correlations between traits is to avoid false positives. If the studied traits are strongly correlated, we cannot say that the associated SNPs are responsible for variation in all those traits, indepently. We cannot ignore that they could rather be just associated with one of these traits, or them in all.

3. Despite the huge genome of pine tree species, I cannot agree that population structure can only be assessed through EST-SSR data. The genome is huge but highly repetitive. Population structure can be assessed using SNP data in selected genes or a higher number of SNPs distributed along the genome. However, population structure assessed with SSRs could be ok. But, if the computed population structure does not represent the structure of the studied population, as said by authors… How could it be used to correct for population structure in the mixed linear model implemented in the association analysis?

4. The statistical methods used for association analysis are still my major concern about the manuscript. I think the authors have not addressed what was highlighted about basic statistical problems which I think are at the base of their results and main findings. If you do not apply a false discovery rate correction method, neither know about genetic correlations between phenotypes in the studied traits, in my opinion, association analysis results are not reliable.

Reviewer #2: All the comments have been address appropriately and manuscript may be accepted for publication. For future publications remember to provide comment by comment response in separate sheet (response sheet) not in the revised manuscript.

7. PLOS authors have the option to publish the peer review history of their article (what does this mean?). If published, this will include your full peer review and any attached files.

Reviewer #1: No

Reviewer #2: No

---

## [Author Response · Author response to Decision Letter 1]

12 Sep 2021

Response Letter

I have received the modification suggestions of two experts, and the modification has been completed according to the expert suggestions. The reply to the expert suggestions is as follows:

Reviewer #1: 

1. Overall the authors tried to address most comments, improving the manuscript. However, I still think that statistical analysis may compromise their findings, and thus their results and conclusions. I’m especially concerned about association testing results, as highlighted in major comment 4. Thus, I do not think this manuscript is suitable for publication in PLOS ONE in its present form. Response to authors for major comments review:

Response: We really appreciate your suggestions that would improve the manuscript. Thus, we have carefully revised the manuscript according to your comments.

2. Authors now provide a citation about heritability for turpentine and alfa-pinene in the species. My sense was that there is a need for trustable genetic control if then you are going to perform an association test, which consists in stablishing an statistic relationship between fenotypes (SNPs) and genotypes.

My interest in testing for genetic correlations between traits is to avoid false positives. If the studied traits are strongly correlated, we cannot say that the associated SNPs are responsible for variation in all those traits, indepently. We cannot ignore that they could rather be just associated with one of these traits, or them in all.

Response: Thank you again for your comments. As you said, if the traits are correlated, it is hard to say that the SNPs are responsible for variation in it. In fact, many studies have shown that there was a significant negative correlation between α-pinene and β-pinene content. That is why I use to different methods when we are choosing plus trees. Before I start this article, we used four genes, including PeTPS-Bpin (β-pinene synthetase gene), not just PeTPS-(-)Apin gene, but no SNPs associated with any of the other traits. Addtionally, candidate gene association analysis is not as accurate as the GWAS method, but the only advantage is to save the cost of the test. We believe that our research has some reference value even though there are shortcomings in technology. I added some correlation data and references in L441-442.

3. Despite the huge genome of pine tree species, I cannot agree that population structure can only be assessed through EST-SSR data. The genome is huge but highly repetitive. Population structure can be assessed using SNP data in selected genes or a higher number of SNPs distributed along the genome. However, population structure assessed with SSRs could be ok. But, if the computed population structure does not represent the structure of the studied population, as said by authors… How could it be used to correct for population structure in the mixed linear model implemented in the association analysis?

Response: Thank you again for your comments. As you said, SNP in selected genes or distributed along the genome can be used to evaluate population structure. However, only a limited number of specific genes have been published in Pinus elliottii, and there were few SNPs in the database. Conversely, there were a large number of SSR markers for P. elliottii itself, as well as general markers for other species of Pinus. The use of SSR markers is also aimed at greatly reducing our research costs. So we selected SSR markers after referring to studies on other Pinus such as Loblolly pine (P. taeda) (Eckert, 2010). Although this method would not perfectly perform an accurate result, it is a reasonable method and has certain reference value. I have made supplementary explanations in the text according to your suggestions (L136-137).

4. The statistical methods used for association analysis are still my major concern about the manuscript. I think the authors have not addressed what was highlighted about basic statistical problems which I think are at the base of their results and main findings. If you do not apply a false discovery rate correction method, neither know about genetic correlations between phenotypes in the studied traits, in my opinion, association analysis results are not reliable.

Response: Thank you very much for your advice. As you said, the results of our previous association analysis are not reliable. I have tested the results of my association analysis with The FDR method as suggested by you, and finally eliminated seven false positives, so that the phenomena I failed to explain before can be scientifically explained. The article has made a large number of modifications (L21-26, 193-195, 312-317, 327, 354-357, 431 ).

 

Reviewer #2: 

All the comments have been address appropriately and manuscript may be accepted for publication. For future publications remember to provide comment by comment response in separate sheet (response sheet) not in the revised manuscript.

Response: Thank you again for your approval of this article.

---

## [Decision Letter · Decision Letter 2]

11 Nov 2021

PONE-D-21-06793R2Association of single nucleotide polymorphisms in the PeTPS-(-)Apin gene with resin traits in Pinus elliottiiPLOS ONE

Dear Dr. Zhang,

Thank you for submitting your manuscript to PLOS ONE. After careful consideration, we feel that it has merit but does not fully meet PLOS ONE’s publication criteria as it currently stands. Therefore, we invite you to submit a revised version of the manuscript that addresses the points raised during the review process.

We look forward to receiving your revised manuscript.

Kind regards,

Himanshu Sharma

Academic Editor

PLOS ONE

Journal Requirements:

Additional Editor Comments (if provided):

Based on reviewers recommendation the manuscript needs revision So authors are required to revise the manuscript thoroughly.

Reviewers' comments:

Reviewer's Responses to Questions

**Comments to the Author**

1. If the authors have adequately addressed your comments raised in a previous round of review and you feel that this manuscript is now acceptable for publication, you may indicate that here to bypass the “Comments to the Author” section, enter your conflict of interest statement in the “Confidential to Editor” section, and submit your "Accept" recommendation.

Reviewer #3: (No Response)

2. Is the manuscript technically sound, and do the data support the conclusions?

Reviewer #3: No

3. Has the statistical analysis been performed appropriately and rigorously? 

Reviewer #3: Yes

4. Have the authors made all data underlying the findings in their manuscript fully available?

Reviewer #3: No

5. Is the manuscript presented in an intelligible fashion and written in standard English?

Reviewer #3: No

6. Review Comments to the Author

Reviewer #3: The work's goal is good, and the authors put forth a lot of effort to convey the findings.

The customized association analysis methodologies, on the other hand, are unreliable.

I don't believe this manuscript is ready for publishing.

There are numerous errors, some of which are as follows:

Correct the spelling ‘TASSLE’ to TASSEL at line 21, 22. Similarly on line 186. Go through the manuscript.

Change ‘rosin’ to resin line 17, 172

Spell out ‘RY’ when you use it first time in the manuscript it is better.

Statements don’t match about the population structure: You write sub-population (K) K1 to K9 in Method section while in K = 2 to 7 is mention in result part.

The method of ∆K was not mention in method section

The value of MCMC reps is also missing

Method is not clear at line 183 – 187 (GC procedure and GC-MS procedure)

Line 188-189: It is structure based association analysis. My main concern is that how could SSR based Q value be used in the MLM model of association mapping which is based on SNPs data. It is difficult to compare population structure data of different molecular marker. You should do structure analysis and generate Q value for SNPs data also. The association analysis data is not reliable.

The used K matrix is based on SSRs data or SNPs data? If this is based on SSRs data again I shocked how you could use this in SNPs based association mapping. You should use SNPs based K matrix as well.

The total no. of sample used in this study is 110 but the sum of all samples from four groups comes 109 in Table 3 line 221

236-237: Nucleotide diversity (0.00276) and watterson’s theta value (0.00259) is low might be due to the less sample size not moderate.

7. PLOS authors have the option to publish the peer review history of their article (what does this mean?). If published, this will include your full peer review and any attached files.

Reviewer #3: No

---

## [Author Response · Author response to Decision Letter 2]

9 Dec 2021

Response to comments and suggestions of reviewers:

1. If the authors have adequately addressed your comments raised in a previous round of review and you feel that this manuscript is now acceptable for publication, you may indicate that here to bypass the “Comments to the Author” section, enter your conflict of interest statement in the “Confidential to Editor” section, and submit your "Accept" recommendation.

Reviewer #3: (No Response)

2. Is the manuscript technically sound, and do the data support the conclusions?

Reviewer #3: No

Response: In this manuscript, 29 resin traits were determined by methods of bark stress wounding and GC-MS, and the sampling and experiment were repeated to ensure accurate and authentic data. Structure V2.3.3 software was used to analyze population structure and genetic relationship, MLM program of TASSEL software was used for SNP association analysis, and FDR program was used to adjust P values. Finally, three SNPs of PeTPS-(-)Apin gene were found to be significantly correlated with α-pinene content, and realistic gain of 118.0 % of α-pinene content was obtained. The research of this manuscript has a certain reference value.

3. Has the statistical analysis been performed appropriately and rigorously?

Reviewer #3: Yes

Response: Thank you very much for your recognition of the statistical analysis method in this manuscript. 

4. Have the authors made all data underlying the findings in their manuscript fully available?

Reviewer #3: No

Response: We have provided woodland photos at 3072×1728 and 3456×4608 resolutions in the support information, along with all raw data summaries and software analysis results. After your reminder, we have checked the uploaded support information. We uploaded the support information S5 as a supplement, and modified the incorrect format of S9, to ensure compliance with the PLOS data policy.

5. Is the manuscript presented in an intelligible fashion and written in standard English?

Reviewer #3: No

Response: The manuscript was edited by Editage. Under your reminding, we checked the manuscript and corrected some mistakes. We returned the manuscript to the Editage's office, re-edited it in English, and uploaded the English editor certificate (filename: Certificate_of_editing.docx) through the submission channel.

 

6. Review Comments to the Author

Reviewer #3: The work's goal is good, and the authors put forth a lot of effort to convey the findings.

The customized association analysis methodologies, on the other hand, are unreliable.

I don't believe this manuscript is ready for publishing.

There are numerous errors, some of which are as follows:

(1) Correct the spelling ‘TASSLE’ to TASSEL at line 21, 22. Similarly on line 186. Go through the manuscript.

Change ‘rosin’ to resin line 17, 172

Response: Thanks for your reminding, and I'm sorry to let you see these careless error. I have checked carefully the whole manuscripts, and corrected these mistakes. In addition, I also corrected other language and grammatical errors through the Editage company.

(1) Lines 21, 198.—“TASSLE” corrected to “TASSEL”.

(2) Lines 16, 17, 35, 37, 94, 173, 306, 306(2), 308, 312, 407, 452 and Table 5 (13 errors in total).—“rosin” corrected to “resin”.

(2) Spell out ‘RY’ when you use it first time in the manuscript it is better.

Response: Thanks for your reminding, and I'm sorry to let you see this careless error. I checked the whole manuscript carefully and corrected all of similar errors.

(1) Lines 157, 168.—“RY”, is not a professional noun abbreviation, and it only appears twice in the manuscript, both corrected to “resin yield”.

(2) Lines 75.— “TPS”, is used for the first time in the manuscript, and corrected to “Terpene synthase (TPS)”.

(3) Lines 97.—“MAS”, is used for the first time in the manuscript, and corrected to “marker-assisted selection (MAS)”.

(4) Lines 382.—“FHA”, is used for the first time in the manuscript, and corrected to “forkhead-associated (FHA)”.

(3) Statements don’t match about the population structure: You write sub-population (K) K1 to K9 in Method section while in K = 2 to 7 is mention in result part.

Response: Thanks for your reminding, and it's my carelessness. I checked the whole manuscript carefully and corrected all of similar errors.

(1) Lines 134.—The K value in the result section is correct, “set K to 1~9” in the material method part is corrected to “K set to 2~7”. 

(2) Lines 133.—There is a parameter value error, and modified “100,000” to correct value “10,000”.

(3) Lines 223.—There is a numerical value error, and modified “37” to the correct value “35”.

(4) The method of ∆K was not mention in method section

The value of MCMC reps is also missing

Method is not clear at line 183 – 187 (GC procedure and GC-MS procedure)

Response: Thanks for your comments. I have added the following contents: 

(1) Lines 134-137.—added the method of optimal K and ΔK value. 

“The optimal value of K was determined using the ΔK method with the online program (Structure Harvester: http://taylor0.biology.ucla.edu/struct_harvest/).” 

(2) Lines 133.—added the parameter of "Number of MCMC Reps after Burnin".

 “"Number of MCMC Reps after Burnin" set to 10,000”.

(3) Lines 183-186.—supplemented the programs of GC and GCMS.

 “GC was carried out using the following parameters: 60 ℃ for 2 min, 5 ℃·min-1 to 80 ℃, 30 ℃·min-1 to 230 ℃, finally 5℃·min-1 to 260 ℃, for 10 min. Injection volume: 0.4 μL. GC-MS was carried out using the following parameters: 60 ℃ for 2 min, 5 ℃·min-1 to 80 ℃, 30 ℃·min-1 to 230 ℃, finally 5℃·min-1 to 260 ℃, for 10 min. Injection volume: 0.1 μL.”.

(5) Line 188-189: It is structure based association analysis. My main concern is that how could SSR based Q value be used in the MLM model of association mapping which is based on SNPs data. It is difficult to compare population structure data of different molecular marker. You should do structure analysis and generate Q value for SNPs data also. The association analysis data is not reliable.

The used K matrix is based on SSRs data or SNPs data? If this is based on SSRs data again I shocked how you could use this in SNPs based association mapping. You should use SNPs based K matrix as well.

Response: Thanks very much for your comments. As you said, it is difficult to compare data of different molecular markers. And we also would like to be able to use more SNP markers distributed along the genome to obtain population structure data. However, it is a pity that this manuscript did not get more SNP markers.

TPS genes were cloned, sequenced, and found SNP loci, and performed a simple correlation analysis (ANOVA) between these SNPs and phenotypic traits. There was a significant association between PeTPS-(-)Apin gene and resin traits. In order to find the association loci suitable for MAS, we tried to add two fixed effects, K-matrix and Q-matrix, into the independent variables of association analysis. However, we obtained only 59 SNPs with an average distance of 78.61 bp, which is difficult to be used for population structure analysis. What is more, it is difficult to obtain more SNPs from the genome in the short term in the case of the P. elliottii genome-wide unknown.

SSR is a broad-spectrum marker distributed throughout the genome and widely used in population structure research. Based on the related research of P. taeda. (Eckert, 2010: https://doi.org/10.1534/genetics.114.164087), we tried to use SSR markers to obtain K matrix and Q matrix. I think the manuscript still has some reference value, even though this method may be not the best.

The discussion section of the manuscript talked about the issues, but it wasn't comprehensive enough. I made some improvements. 

Line 347-353.— The original: “We still cannot rule out the possibility of false positives in the association analysis because of the K matrix and Q matrix were calculated using SSR markers, since the available SNP markers in the association analysis of candidate genes could did not cover the whole genome. Nevertheless, some information can be obtained from the limited data. ” 

The revised: “Based on a small data volume of nucleotides database (less than 300 ), our existing SNP data cannot cover the genome-wide. Moreover, it is difficult to obtain more SNP data in the short term due to the enormous genome (>20,000 Mbp). Therefore, the K-matrix and q-matrix in association analysis are calculated using SSR markers, which may be controversial. However, some information can be obtained from the limited data.”

(6) The total no. of sample used in this study is 110 but the sum of all samples from four groups comes 109 in Table 3 line 221

Response: Thanks very much for your reminding. There were 110 samples in the manuscript. But in subgroups grouping, we classified the sample with proportion of a single color exceeded 40 % (Q≥0.4) into the subpopulation of that color. However, the maximum Q value of sample No.59 was less than 0.4, which could not meet the grouping conditions. Therefore, there were only 109 samples in Table 3.

I supplemented the method of subgroups grouping and the description of related problems:

Line 220-222 : “When the proportion of a single color exceeded 40 % (Q≥0.4), we classified the individual into the subpopulation of that color, but the maximum Q value of sample No. 59 was less than 0.4, so no statistics were collected. ”

(7) 236-237: Nucleotide diversity (0.00276) and watterson’s theta value (0.00259) is low might be due to the less sample size not moderate.

Response: I agree with you and revise the manuscript accordingly. 

Line 253-255: “The nucleotide polymorphism of π 0.00276 and θw 0.00259, were at a low level, probably because more polymorphisms were not detected from the small sample size in this study.”.

---

## [Decision Letter · Decision Letter 3]

2 Feb 2022

PONE-D-21-06793R3Association of single nucleotide polymorphisms in the PeTPS-(-)Apin gene with resin traits in Pinus elliottiiPLOS ONE

Dear Dr. Zhang,

Thank you for submitting your manuscript to PLOS ONE. After careful consideration, we feel that it has merit but does not fully meet PLOS ONE’s publication criteria as it currently stands. Therefore, we invite you to submit a revised version of the manuscript that addresses the points raised during the review process.

The manuscript entitled "Association of single nucleotide polymorphisms in the PeTPS-(-)Apin gene with resin traits in Pinus elliottii " by Lu Zhang, is not acceptable in the present form and one of the reviewer again has some queries So manuscript again needs to be revised.

We look forward to receiving your revised manuscript.

Kind regards,

Himanshu Sharma

Academic Editor

PLOS ONE

Journal Requirements:

Additional Editor Comments:

The manuscript entitled "Association of single nucleotide polymorphisms in the PeTPS-(-)Apin gene with resin traits in Pinus elliottii " by Lu Zhang, is not acceptable in the present form and one of the reviewer again has some queries So manuscript again needs to be revised.

Reviewers' comments:

Reviewer's Responses to Questions

**Comments to the Author**

1. If the authors have adequately addressed your comments raised in a previous round of review and you feel that this manuscript is now acceptable for publication, you may indicate that here to bypass the “Comments to the Author” section, enter your conflict of interest statement in the “Confidential to Editor” section, and submit your "Accept" recommendation.

Reviewer #3: (No Response)

2. Is the manuscript technically sound, and do the data support the conclusions?

Reviewer #3: Partly

3. Has the statistical analysis been performed appropriately and rigorously? 

Reviewer #2: Yes

Reviewer #3: Yes

4. Have the authors made all data underlying the findings in their manuscript fully available?

Reviewer #3: Yes

5. Is the manuscript presented in an intelligible fashion and written in standard English?

Reviewer #3: Yes

6. Review Comments to the Author

Reviewer #3: Dear Editor,

Most of the writing is now ok, and the authors’ provides substantial information. But again my main concern is about the association mapping approaches adapted by the authors in the manuscript. Authors used Q value of SSRs in SNP based association mapping. It is unacceptable as the population structure reduced the biasness of association by clustering of genotypes on the basis of polymorphic/unlinked marker irrespective of other variables. It is difficult to relate the location of SSRs and SNPs marker (linked or unlinked). In this paper there is a chance of highly wrong association. The genetic proportion value of admixture level will be totally different with different molecular marker if the nature of marker is different. If Q value of SSR is used in SNPs based association analysis it gives false association. The association analysis based on ANOVA is ok moreover the population structure /diversity analysis of P. elliotti based on SSR data is ok but Q matrix and K matrix generated through SSRs data cannot be used in SNPs based association analysis it can only be used when the physical location of the both markers is within the LD range. Have you compare the physical location of those SSRs to the SNPs? In my suggestion, population structure based association data should be remove from the manuscript and revised the manuscript accordingly.

7. PLOS authors have the option to publish the peer review history of their article (what does this mean?). If published, this will include your full peer review and any attached files.

Reviewer #3: No

---

## [Author Response · Author response to Decision Letter 3]

11 Mar 2022

Response to comments and suggestions of reviewers:

1. If the authors have adequately addressed your comments raised in a previous round of review and you feel that this manuscript is now acceptable for publication, you may indicate that here to bypass the “Comments to the Author” section, enter your conflict of interest statement in the “Confidential to Editor” section, and submit your "Accept" recommendation.

Reviewer #3: (No Response)

2. Is the manuscript technically sound, and do the data support the conclusions?

Reviewer #3: Partly

Response: Thanks for your recognition. We deleted the association data based on population structure in accordance with your requirement, and focused on SNP analysis of PeTPS-(-)Apin gene. Genotyping of the population based on three SNPs, and the plus trees with high α-pinene content was selected. Finally, the actual α-pinene content was 118.0%. The research of this paper has certain reference value.

3. Has the statistical analysis been performed appropriately and rigorously?

Reviewer #3: Yes

Response: Thank you very much for your recognition of the statistical analysis method in this manuscript. 

4. Have the authors made all data underlying the findings in their manuscript fully available?

Reviewer #3: Yes

Response: Thank you very much for your recognition of the data underlying the findings in this manuscript.

5. Is the manuscript presented in an intelligible fashion and written in standard English?

Reviewer #3: Yes

Response: Thank you very much for your recognition.

 

6. Review Comments to the Author

Reviewer #3: Dear Editor,

Most of the writing is now ok, and the authors’ provides substantial information. But again my main concern is about the association mapping approaches adapted by the authors in the manuscript. Authors used Q value of SSRs in SNP based association mapping. It is unacceptable as the population structure reduced the biasness of association by clustering of genotypes on the basis of polymorphic/unlinked marker irrespective of other variables. It is difficult to relate the location of SSRs and SNPs marker (linked or unlinked). In this paper there is a chance of highly wrong association. The genetic proportion value of admixture level will be totally different with different molecular marker if the nature of marker is different. If Q value of SSR is used in SNPs based association analysis it gives false association. The association analysis based on ANOVA is ok moreover the population structure /diversity analysis of P. elliotti based on SSR data is ok but Q matrix and K matrix generated through SSRs data cannot be used in SNPs based association analysis it can only be used when the physical location of the both markers is within the LD range. Have you compare the physical location of those SSRs to the SNPs? In my suggestion, population structure based association data should be remove from the manuscript and revised the manuscript accordingly.

Response: Thank you very much for your comments and suggestions, and I agree with you. 

According to your suggestion, I deleted the section of association analysis based on population structure. I changed the title to "Analysis on single nucleotide polymorphisms of the PeTPS-(-)Apin gene in Pinus Elliottii". The focus of this manuscript was changed to SNPs analysis of PeTPS-(-)Apin gene, avoiding highly wrong association. However, due to the particularity of the samples, I still retained the contents of phenotypic determination and population structure analysis in the form of supporting information. 

Changes to the framework of the article were as follows:

(1) Lines 126,178, “Materials and methods” section — “Population structure analysis”, “Determination of resin production capacity”, “Determination of turpentine composition”, and “Association analysis” section was completely deleted. At the same time, added support information files S3 (Determination method for resin traits) and S5 (Group structure analysis method and results).

(2) Lines 128,139. “Materials and methods” section — The section “PeTPS-(-)Apin gene cloning and SNP site typing” was divided into two parts “PeTPS-(-)Apin gene cloning” and “SNP site analysis and typing”.

(3) Line 179. “Results” section — “Group structure and kinship” section was completely deleted. And added it to the support information files S5 (Text. Group structure analysis method and results).

(4) Lines 182,376. “Results” section — The section “PeTPS-(-)Apin gene sequence and SNP sites” was divided into two parts “PeTPS-(-)Apin gene sequence” and “Diversity analysis of SNPs”.

(5) Lines 371,389. “Results” section — The section “Linkage disequilibrium and haplotype block” was divided into two parts “Linkage disequilibrium” and “Haplotype block”.

(6) Line 409. “Results” section — Change the section “Resin traits and SNP association analysis” to “Genotyping of SNPs association with resin traits”.

Other corresponding modifications are as follows: 

(1) Lines 22-24. “Abstract - Methods” section — Change “Bark stress wounding and GC-MS were used to determine 29 resin traits. Structure v2.3.3 software was used to analyze population structure and genetic relationships, DnaSP v4.0 software was used to evaluate genetic diversity, the MLM program of TASSEL was used for SNP association analysis, and false discovery rate (FDR) was used for adjusting the p value.” 

to “PeTPS-(-)Apin gene was cloned by double primers (external and internal). DnaSP V4.0 software was used to evaluate genetic diversity and linkage disequilibrium. SHEsis was used for haplotype analysis. SPSS was used for ANOVA and χ2 test.”.

(2) Lines 98-100. “Introduction” section — Change “the PeTPS-(-)Apin gene was cloned, 29 resin traits were determined, and functional single nucleotide polymorphisms (SNPs) were screened by association analysis. The optimal selection scheme of P. elliottii with high α-pinene content was also considered.” 

to “the PeTPS-(-)Apin gene was cloned, and its single nucleotide polymorphisms (SNPs) and linkage disequilibrium were analyzed. Functional SNPs were screened by ANOVA, 110 samples were typed, and the optimal selection scheme of P. elliottii with high α-pinene content was also considered.”.

(3) Lines 178-179. “Materials and methods” section — Added the sentence “SHEsis was used for haplotype analysis. ANOVA and χ2 were performed to evaluate if the results conformed to the Hardy-Weinberg equilibrium using SPSS software.”.

(4) Line 423. “Results - Genotyping of SNPs association with resin traits” section — The original Table 5 was deleted, and added it to Schedule 1.

(5) Line 433. “Results - Genotyping of SNPs association with resin traits” section — A new table 5 was added “Genotypes frequency and polymorphisms of 110 Pinus elliottii samples based on 3 SNPs”.

(6) Line 422. “Results - Genotyping of SNPs association with resin traits” section — Added the sentence “The determination methods of these resin traits were shown in supporting information (S3, S4).”.

(7) Lines 426-428. “Results - Genotyping of SNPs association with resin traits” section — Added the sentence “Genotyping and polymorphism analysis were performed on 110 samples of Pinus elliottii. The expected heterozygosity, had an average of 0.5365, and the observed heterozygosity, had an average of 0.5030 (Table 5).”.

(8) Lines 474-477. “Discussion” section — Change “Therefore, the K-matrix and q-matrix in association analysis are calculated using SSR markers, which may be controversial. Nevertheless, some information can be obtained from the limited data. Using candidate gene association analysis, we divided the PeTPS-(-)Apin gene sequence into six haplotypes (Fig 7), screened out 3 loci associated with α-pinene content from 59 SNPs, and performed polymorphism analysis (Table 7). The expected heterozygosity, had an average of 0.5365, and the observed heterozygosity, had an average of 0.5030. The linkage disequilibrium r2 values of the TagSNPs are listed in Table 8. There was a strong linkage between CG615 and AT641 in haplotype block (-)Apin-1 (r2=1). We found that the contribution rate of the PeTPS-(-)Apin gene to α-pinene content was 37.264 %; thus it was considered an important candidate gene for this trait.” 

to “Therefore, simple associations (ANOVA) were used for association analysis, which may be controversial. Nevertheless, PeTPS-(-)Apin gene was considered as an important candidate gene for α -pinene content, and three TagSNPs (CG615, AT641 and AG3859) were associated with α -pinene content.”.

(9) Line 567. “Discussion” section — “Among the 12 trees, 2 (16.67%) were from the Red group, 3 (25%) are from the Green group, and 7 (58.33%) are from the Blue group.” was deleted.

---

## [Decision Letter · Decision Letter 4]

23 Mar 2022

Analysis on Single Nucleotide Polymorphisms of the PeTPS-(-)Apin Gene in Pinus elliottii

PONE-D-21-06793R4

Dear Dr. Zhang,

We’re pleased to inform you that your manuscript has been judged scientifically suitable for publication and will be formally accepted for publication once it meets all outstanding technical requirements.

Kind regards,

Himanshu Sharma

Academic Editor

PLOS ONE

Additional Editor Comments (optional):

The manuscript entitled Analysis on Single Nucleotide Polymorphisms of the PeTPS-(-)Apin Gene in Pinus elliottii is extensively revised by the authors and answered all the queries by each reviewer.

There are always chances of improvement like correction in grammatical errorrs and many others, which can be corrected at the time of revision.

Reviewers' comments:

Reviewer's Responses to Questions

**Comments to the Author**

1. If the authors have adequately addressed your comments raised in a previous round of review and you feel that this manuscript is now acceptable for publication, you may indicate that here to bypass the “Comments to the Author” section, enter your conflict of interest statement in the “Confidential to Editor” section, and submit your "Accept" recommendation.

Reviewer #2: All comments have been addressed

Reviewer #3: All comments have been addressed

2. Is the manuscript technically sound, and do the data support the conclusions?

Reviewer #2: Yes

Reviewer #3: Yes

3. Has the statistical analysis been performed appropriately and rigorously? 

Reviewer #2: Yes

Reviewer #3: Yes

4. Have the authors made all data underlying the findings in their manuscript fully available?

Reviewer #2: Yes

Reviewer #3: Yes

5. Is the manuscript presented in an intelligible fashion and written in standard English?

Reviewer #2: Yes

Reviewer #3: Yes

6. Review Comments to the Author

Reviewer #2: The paper entitled "Analysis on Single Nucleotide Polymorphisms of the PeTPS-(-) Apin Gene in Pinus elliottii" may be accepted for publications as all the comments raised were addressed.

Reviewer #3: Dear Editor,

Authors provided substantial information and extensively revised the manuscript. Now it can be accepted for publication.

Minor correction

Sentence "MLM program of TASSEL was used for SNP association analysis

, and false discovery rate ( FDR) was used for adjusting the p value" should be removed/corrected from abstract. As the revised manuscript described the association analysis result based on ANOVA.

7. PLOS authors have the option to publish the peer review history of their article (what does this mean?). If published, this will include your full peer review and any attached files.

Reviewer #2: No

Reviewer #3: No

---

## [Editor Report · Acceptance letter]

5 Apr 2022

PONE-D-21-06793R4 

Analysis on Single Nucleotide Polymorphisms of the *PeTPS-(-)Apin* Gene in *Pinus elliottii*

Dear Dr. Zhang:

I'm pleased to inform you that your manuscript has been deemed suitable for publication in PLOS ONE. Congratulations! Your manuscript is now with our production department. 

Kind regards, 

on behalf of

Dr. Himanshu Sharma 

Academic Editor

PLOS ONE